# Crossing the Power Line: Using Virtual Simulation to Prepare the First Responders of Utility Linemen

**Alaina Herrington** [1,*] and **Joseph Tacy** [2]

1   Judith Gore Gearhart Clinical Skills Center, University of Mississippi, Jackson, MS 39216, USA
2   School of Nursing, University of Mississippi, Jackson, MS 39216, USA; jtacy@umc.edu
*   Correspondence: aherrington@umc.edu

**Abstract:** Virtual reality (VR) healthcare simulation has helped learners develop skills that are transferable to real-word conditions. Innovative strategies are needed to train workers to improve community safety. The purpose of this pilot project was to evaluate the use of a VR simulation applying the International Nursing Association for Clinical Simulation and Learning (INACSL) Standards of Best Practice: Simulation[SM] Simulation Design with eight power line workers. Six power industry supervisors and educators assisted in facilitating three VR simulations with eight linemen participants. Kotter's eight steps to leading change and the INACSL Standards of Best Practice: Simulation[SM] Simulation Design were utilized in working with energy leaders and VR developers to carry out this pilot project. Pre- and post-implementation surveys demonstrated a 28% improvement in participants' learning outcomes. All three learning objectives were met. This project demonstrated the successful application of a translational framework and the INACSL Standards of Best Practice: Simulation[SM] in a VR context in the power industry. This process may be helpful to guide or inspire further adoption of VR in unconventional settings.

**Keywords:** virtual reality; simulation best practices; nonhealthcare training

## 1. Introduction

There is overwhelming evidence that virtual reality (VR) healthcare simulation increases learning outcomes [1,2]. VR simulations have recently been at the forefront of helping to increase the number of healthcare providers prepared to fight the COVID 19 virus, making VR an invaluable asset in education [3]. Though healthcare educators had a plan B to provide much-needed just-in-time training—that of VR—other disciplines may not have been as prepared to train virtually. Although the use of VR simulation is common in healthcare, VR can reach beyond the healthcare discipline to influence safety throughout the community. This article will discuss how one simulation nurse educator utilized Kotter's Change Theory and the International Nursing Association for Clinical Simulation and Learning INACSL Standards of Best Practice: Simulation[SM] Simulation Design [4] to share simulation best practices in the context of the power industry to improve skills acquisition for utility workers who work in the community.

The term "first responder" refers to those individuals who in the early stages of an incident are responsible for the protection and preservation of life, property, evidence, and the environment, including emergency response providers as defined in by Homeland Security Act of 2002 (6 U.S.C. § 101), as well as emergency management, public health, clinical care, public works, and other skilled support personnel (such as equipment operators) that provide immediate support services during prevention, response, and recovery operations [5]. During a large-scale natural disaster, emergency first responders are often the first to pick up the pieces in the aftermath. Although utility linemen are often not considered as first responders, they have one of the most dangerous jobs in the world. These men and

women venture into war-torn circumstances to handle high-powered electrical lines. Many of these downed power lines are tangled with fallen trees and other debris, and the linemen are often working in the dark, making it that much more dangerous. On average around 19.2 workers per 100,000 are killed on the job every year—twice more than the fatality rate of police officers and firemen [6]. Further, many other linemen suffer loss of limbs from electrical burns and mechanical trauma.

Part of one Southeastern state simulation center's mission is to enhance community outreach. To meet this goal, and to possibly create a revenue stream, one-simulation nurse educator worked with energy leaders and VR developers to create a pilot training program designed to provide a "hands-on" learning experience for the energy workforce where identified energy-related challenges have been noted. This VR experience engaged learners through an on-screen utility line environment that duplicated real-life senses, thinking, and behaviors in a 3D immersive simulation wearing Oculus Go devices. This virtual sense of spatial presence made learners feel like they were in real time at a utility pole, thus providing a standardized method to train large numbers of the geographically diverse energy workforce throughout a multistate region. The INACSL Standards of Best Practice: Simulation$^{SM}$ are designed to guide healthcare simulations globally, although the first author decided to apply them uniquely to the power industry. There is current evidence that the INACSL Standards of Best Practice: Simulation$^{SM}$ Simulation Design have been successful in creating effective virtual reality simulations for nurses [7–11]. Some of the lessons learned in developing a Neonatal Intensive Care evacuation virtual story board were to involve stakeholders in the development of objectives, description of the scene, and identification of realistic opportunities [7]. Nurse educators have also utilized the standards to teach resuscitation in a virtual simulation game [8]. These educators found the platform needed to be filmed from the participants' perspective to provide realism and an immersive experience. They utilized the learning objectives to make focused, preselected decision points during the encounter. Nurse educators have also used VR to enable learners to associate classroom learning with clinical learning regarding chronic care management in community-based settings [9]. These educators found it was critical that the facilitator had adequate simulation education to be effective at teaching with this modality. Nurse educators found virtual simulation nurtures immersive and problem-based learning experiences for nursing students to exercise decision-making and utilizing the evidence [10]. Nurse educators in Korea found that despite the difficulties with implementing VR simulation, the possible benefits offset the challenges [11]. They felt the utilization of VR offers nurses opportunities to cooperate with interdisciplinary teams, and enlarge scientific data into practice. Based on these findings and many others, the INACSL Standards of Simulation Best Practice have been foundational in guiding robust, quality healthcare simulations. The purpose of this pilot project was to evaluate use of a VR simulation applying the International Nursing Association for Clinical Simulation and Learning (INACSL) Standards of Best Practice: Simulation$^{SM}$ Simulation Design with power line workers.

## 2. Methods

This project's goal was to translate existing nursing evidence of effectively using the INACSL Standards of Simulation Best Practices to nonhealthcare workers and conduct a program evaluation. Six energy supervisors and educators assisted in facilitating and evaluating eight linemen participants with varied experiences in three simulations. Kotter's (1996) eight steps to leading change were utilized in carrying out this project [12] (Figure 1). Kotter's steps provided a practical manner for the simulation nurse educator to implement the new training. (1) The educator created a sense of urgency by reminding the planning team of the deadline for grant funding to get the group motivated to begin the project. (2) The educator gathered a group to create an activity planning team. The team consisted of the simulation nurse educator, an energy manager, an energy engineer, an innovation officer, and a VR developer. (3) A common vision of the project was created at the first meeting. Tasks were identified and assigned to a team member. (4) The energy manager was responsible to identify a group that would benefit the most from piloting the simulations. (5) The energy managers

and supervisors had minimum simulation and debriefing knowledge. The simulation nurse educator trained these individuals before the pilot project to ensure best practices were implemented. (6) The simulation results were shared in an energy executive regional meeting to celebrate the short-term accomplishment. (7) In order to sustain the simulation training, the energy manager requested a new budget line item be added for future simulation trainings. (8) To create institutional change, simulation surveys are now being conducted to capture related practice changes.

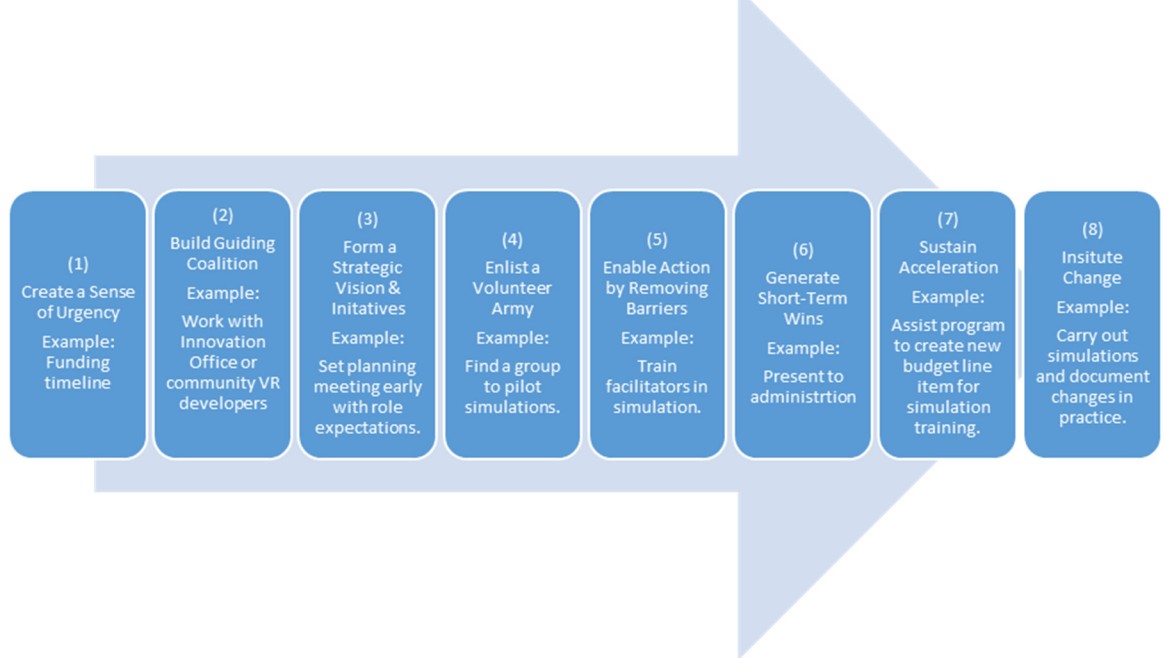

**Figure 1.** Kotter's steps applied to crossing the power line to implement simulation training with a diverse workforce.

The INACSL Standards of Best Practice: Simulation[SM] were used to guide the project. Specifically, the Standard of Simulation Design provided specific guidance in the scenario development and implementation [8]. The following breakdown highlights how the criteria within the Standard were utilized.

Criterion 1: Perform a needs assessment to provide the foundational evidence of the need for a well-designed simulation-based experience. The simulation nurse educator identified the learning gap by asking ten energy experts from across the state eight open-ended questions. Participant responses to the open-ended questions were read, and traditional thematic analysis techniques were applied individually by the simulation nurse educator and then a second individual. The themes were reviewed by both and a consensus reached on the final results. The experts reported the current state of training was being conducted by multiple energy trainers. Because these trainers come from diverse backgrounds and experiences, current participants in the training did not have the same learning opportunities. There was not a standardized guideline to reference for the training. The current training methods required crews to travel to multiple sites to assess various utility pole issues. Additionally, the utility company desired to become more proactive rather than reactive to prevent future outages.

Criterion 2: Construct measurable objectives. The specific objectives were developed by the planning team at one of the early planning meetings. The team ensured the objectives would meet the larger organizational goals. The objectives were:

1.　Demonstrate competency of reliability skills by identifying at least five damaged sections of the utility pole;

2. Recommend three repair procedures to the utility pole to prevent future outages;
3. Identify four scene safety hazards.

Criterion 3: Structure the format of a simulation based on the purpose, theory, and modality, for the simulation-based experience. (Purpose) The purpose of the simulations was to provide specific actionable feedback for learner improvement and reflection. (Theory) The active learning theory, constructivism, was utilized to ensure the learners' ability to acquire problem-solving skills that cannot be taught and must be learned. (Modality) Considering the need to immerse the learner into the surrounding areas (traffic, pedestrians, and other scene safety hazards) around the utility pole, a VR modality was selected to best meet the learning objectives. Energy supervisors toured the VR developers' office to decide which VR platform would best fit their needs. With the limited grant budget of $25,000, Oculus Go's (Facebook Technologies, Menlo Park, CA, USA) were purchased (at approximately $5500 USD). The VR developer charged $10,000 to develop the virtual simulation content. The simulation educator's time was calculated to be $8000. Additionally, travel and supplies were projected to be approximately $1000.

An energy expert educator worked with the simulation expert educator to create the simulation templates and timing of events for each scenario. These experts utilized current energy protocols to script the cases. The intent was to ensure that the simulation engaged the learner to demonstrate the specific knowledge, skills, or attitudes that need to be assessed. The energy educator revealed the need for the participant to be able to see images both close and in wide-angled images. The specific objectives guided the VR developing team during filming to ensure the correct damaged sections of the utility pole were captured for the simulation.

Criterion 4: Design a scenario or case to provide the context for the simulation-based experience. For the participant to meet the learning objectives, the simulation video needed to be at least ten minutes long at each scene. In the first two simulations, participants wore the Oculus Go's and were virtually at a busy downtown location viewing multiple utility pole issues. They were tasked to communicate with one another and identify line arrestors that can cause outages on the mainline, broken and/or cracked insulators, grounds inside a primary zone that need to be removed, spacing issues, and guy wires in the primary zone without guy strain insulators. Additionally, participants recognized issues related to broken poles, cross arms in distress, and failing equipment. In the third simulation, participants wore the Oculus Go's and were tasked to back up a bucket truck using predetermined hand signals to their driver while performing an area safety assessment before executing aerial tasks. A child riding a bike was present in the scene as a safety hazard.

The participants were given a verbal report of why they had reported to the scene and assigned as either an experienced lineman or a new lineman. Having two participants, instead of the normal one lineman on site, enabled the two participants to communicate during the simulation, and the facilitator to hear the linemen's thought process to see if the learning objectives were met. Additionally, moving cues related to scene safety were placed in the video. For example, in one of the simulations, the safety hazard present was a child riding his bike in the repair area. One of the participants had been assigned to assist with backing up a utility truck to the repair site. The energy educator knew some participants may not look behind them during the simulation to realize there was a safety hazard. To ensure this objective was met, a child's laugh was played to cue the participant of the safety hazard.

Criterion 5: Use various types of fidelity to create the required perception of realism. Active participants were immersed in the simulation through normal scene sounds like car noises, people talking, and wind. Participants were asked to wear their safety gear such as a vest and hard hats during the simulation. Participants were encouraged to carry out the motions as they were there in real life. For example, they demonstrated hand signals as they backed the bucket truck up to the site. Observers were able to see the same scene projected on a large screen television during the simulation. The room the simulations were conducted with enough space for the participant to move around during the virtual experience without getting hurt.

Criterion 6: Maintain a facilitative approach that is participant-centered and driven by the objectives, participant's knowledge or level of experience, and the expected outcomes. In order to ensure the facilitators remained participant-centered and allowed the objectives to drive the learning activity, the simulation educator conducted a train the trainer 4 h course. The six energy supervisors and educators learned how to consider each learner's learning level and the benefits of allowing the learner to learn from their environment and possibly their mistakes. The simulation educator demonstrated how the scenario objectives drive the cues, timing, props, debriefing, and evaluation tools to the new facilitators. The new facilitators also learned how to assess the learners' expected outcomes and provide bidirectional feedback centered around the participants' actions that occurred in the simulation.

Criterion 7: Begin simulation-based experiences with a prebriefing. Before the simulation, the energy facilitators discussed the learning objectives, role expectations, limitations, and fictional contract with the participants. Each participant and facilitator completed a confidentiality form. The facilitators detailed how the VR headset would work and where the participant would be standing in the room to complete the simulation. Participants were asked if they have a history of motion sickness and oriented to the large space where they were to move around during the simulation to prevent fall hazards. The facilitator assured the participants they would be monitoring their location in the room at all times to prevent possible safety issues.

Criterion 8: Follow simulation-based experiences with a debriefing and/or feedback session. The simulation nurse educator utilized the Structured and Supported Debriefing Theory to train the facilitators before the simulation event [13]. The operational acronym for this theory is GAS, which stands for gather, analyze, and summarize. This theory was developed in collaboration with the American Heart Association. A 30-min debriefing was conducted by an energy facilitator after each simulation. All eight participants were included in the debriefing by identifying the issues they discovered and their rationale. The simulation educator assisted the energy facilitator in moving the group through the different stages of the debriefing.

Criterion 9: Include an evaluation of the participant(s), facilitator(s), the simulation-based experience, the facility, and the support team. A "backwards design" process was primarily used during the instructional design phase to create the assessment checklists/rubrics. The simulation educator utilized current energy protocols and worked with the energy supervisors to ensure the tools were mapped to competencies and objectives. The tools utilized in this project were borrowed from other current simulation projects that the simulation educator grounded in simulation evidence-based practices. These tools have been previously tested with healthcare students and evaluated by using performance metrics to analyze the assessment tools. The tools were reviewed and minimally modified to fit this specific project assessment goals.

The quantitative data were analyzed using IBM SPSS Statistics, Version 26 (IBM, Armonk, NY, USA). The Confidence Self Evaluation Tool, which consisted of a 7-item questionnaire with a Likert scale, was utilized to assess confidence pre and post implementation. A paired samples t-test was conducted to compare the overall sum score of the survey tool. A participant satisfaction rubric was collected from the study participants as well as a facilitator validation rubric that was collected from the instructors. Mean score data analysis was performed for both rubrics.

The participants' pre- and post-confidence surveys, participants' simulation satisfaction surveys, and the facilitators' scenario validation surveys were utilized to obtain insight regarding if power line workers' outcomes are effectively met after applying the INACSL Standards of Best Practice: Simulation[SM] Simulation Design in VR simulations. The Participant Pre- and Post- Confidence Self Evaluation Tool was developed by the planning team using the correct participant actions for each scenario. The survey consisted of seven items on a five-point Likert scale (A) "Lack confidence", (B) "I have low confidence", (C) "Neither", (D) "I am confident', or (E) "I am very confident". The same tool was completed by each participant before and after the simulation to assess the difference in confidence after the simulation of the participant in inspecting insulators, tie wires, dead-end bells,

conductors, and jumpers. The survey also asked how confident the participant was at identifying area hazards, cause of damage to a pole, and on making recommendations of repair to a pole.

The Participant Satisfaction Rubric consisted of five "yes" or "no" items assessing if the objectives were met, if the content was relevant, if mastery of the skill was enhanced, if the quality of instruction was excellent, and if the instructor utilized time efficiently. Two open-ended questions were also asked to find out the strengths and weaknesses of the VR training over traditional training methods.

The Facilitator Validation Rubric consisted of seven "yes" or "no" questions assessing if the content matched the learner's current skill level; if the content was accurate, current, and relevant; if appropriate cues were used; if critical thinking and decision making was supported; if adequate time was allocated for decision making; if equipment and preparation of the event supported the learning objectives; and if critical events were supported by the debriefing.

Criterion 10: Provide preparation materials and resources to promote participants' ability to meet identified objectives and achieve expected outcomes of the simulation-based experience. Prior to participants rotating through multiple VR simulations, the participants were asked to review prerecorded video trainings of common issues seen with dysfunctional utility poles and scene safety.

Criterion 11: Pilot test simulation-based experiences before full implementation. This project served as a pilot project to test and see if the training could meet the learning objectives.

## 3. Results

A mix of active participants was chosen, including one new lineman, a lineman with eight months of experience, and two experienced linemen. All four active participants were men in age range from 22–63. Four additional participants who served in an observational capacity (3 men and 1 woman) were chosen, including an operation coordinator, line supervisor, engineer, and engineer supervisor. Their age ranged from 31–55. Before the simulation, the planning group developed multiple evaluation methods.

Eight of the participants participated in the Participant Pre- and Post- Confidence Self Evaluation Tool. Pre-and post-implementation surveys documented a positive delta mean of 100% participant improvement from "I am confident" to "I am very confident" in inspecting dead-end bells. There was a 50%mean improvement of participants from "I am confident" to "I am very confident" in identifying the cause of damage to a pole and making recommendations of repair. There was 0% mean improvement in inspecting insulators, tie wires, and conductor/jumpers, and in identifying area hazards. A paired-samples t-test was conducted to compare the overall sum score of the Pre- and Post-Confidence Self Evaluation Tool. There was a significant difference in the scores for pre- (M = 28, SD = 0.00) to post- (M = 30, SD = 1.07); t (7) = −5.292, $p$ = 0.001. This represents an overall increase in confidence of participants of the VR simulation event.

The Participant Satisfaction Rubric revealed 100% of participants felt the learning objectives were met, the content of the session was relevant to their needs, the session effectively enhanced their mastery of a skill, the quality of instruction was excellent, and the instructor used training time efficiently. Participants listed several areas of strength in the training, as the sights and sounds were accurate; training provided a modality to train a large number of people to increase safety, training helped to teach less skilled to identify problems, training avoided the risk of harm in the field, training brought reality into play, training was similar to the real-world training, training saved time of having to go to the field, and the training provided the ability to discuss issues in a group setting. Participants also provided some areas of improvement for the training, such as the training should provide more movement by the controller, the video should have mostly zoomed-in clearer images, and more case content is needed in the orientation.

The Facilitator Validation Rubric revealed that 100% of instructors that rated the case content matched the participant's current skill level; case content was accurate, current, and relevant; critical thinking and decision making were supported; adequate time was allocated for decision making; the equipment and preparation of the training supported participant objectives; and critical

events were supported by the debriefing. A mean of 17% of instructors felt the cues and consequences could be improved to signal the participant more during the simulation.

## 4. Discussion

The results of this pilot project demonstrated the successful application of a translational framework and the INACSL Standards of Best Practice: Simulation[SM] in a VR context in the power industry. The learning objectives of nonhealthcare workers were met by applying the INACSL Standards of Best Practice: Simulation[SM] Simulation Design in VR simulations. It is important to note that this project only focused on one of the INACSL Standards, as there are many standards. The facilitators and participants both indicated this type of training was effective, and only minor revisions of cues, visual clarity, and orientation to the simulation were suggested to expand this training in the future. To improve these cues, the VR developers plan to reshoot the virtual area of the simulation with a higher resolution camera to enable the learner to see more details of the equipment closer up. Additional scripted details will be added to the scenario template to immerse the learner in the background of the simulation event to cue the participant more on what tasks need to be accomplished. Although these items of improvement were needed, the participants and facilitators perceived that learning took place virtually.

Through this VR learning experience, participants' knowledge was improved. The energy organization plans to use feedback from this training to improve multiple areas of safety, staffing, and training location restrictions within their organization. The participants felt this type of training would also help with improving competencies. Recently, the energy organization made a new change to their policy requiring that every lineman performing a skill on a job site have six years of energy experience at the company to prevent safety issues from occurring. Many of these linemen have over fifteen years of lineman experience and are qualified to complete these tasks, but under the new policy, these individuals cannot complete tasks on-site anymore because they have been with their current organization less than six years. VR simulations will provide the energy company with not only a training platform but also an avenue for an experienced lineman to demonstrate their competency. This change is especially important, because over 40% of the energy lineman will be eligible for retirement in the next few years.

The next steps are to engage surrounding states' energy executives in the VR training sessions. Through this platform, one trainer can lead through a long-range distance platform. The possibility of debriefing in the Metaverse is being discussed to enable the trainer to remotely debrief.

### 4.1. Recommendations

The simulation nurse educator learned multiple lessons using Kotter's framework when working with a diverse workforce and VR developers. Planning early and assigning roles for the project was critical to the project's success. Learning objectives are not commonly utilized in other discipline's training programs. Creating a shared mental model of the learning objectives helped to clarify the simulation design and desired learning outcomes. Identification of a simulation champion and small group motivated to complete the pilot created positive momentum. A focused foundational four-hour simulation course for the energy facilitators initiated the groundwork needed to create a safe, effective learning environment for the learners. Without this knowledge, facilitators revealed they would not have implemented advocacy and inquiry style questioning and would have reverted to their typical closed-ended questioning when discussing participants' performance. Generating, sustaining, and changing the organizational training structure were the longest stages of the process. Obtaining a presentation time and budget line items for the trainings from administration took perseverance. This step was only achieved by involving a strong simulation champion within the organization with influence on key administrators. This framework can be used by other simulation educators seeking to assist programs outside of healthcare to implement best practices into VR simulations.

*4.2. Benefits*

There are many benefits for leaders in simulation programs to consider when partnering with community organizations. The first benefit is to enable the simulation centers to meet their community mission. Not all simulation centers include improving education and safety in the community in their mission; however, simulationists might want to consider sharing their expertise as they know virtual simulations are a reliable method to achieving skills-based competencies [1]. One unexpected benefit of partnering with a nonhealthcare partner is learning the safety practices of other disciplines. Simulation researchers have acknowledged multiple times that knowledge is richer when disciplines learn from each other [14,15]. In this project, the simulation educator learned from the energy educators about how to improve documentation and safety of a hazardous scene. The energy organization utilized a Job Hazard Analysis Form, which required a leader to document the following: hazards, solutions to hazards, a potential rally point, assigned observer of actions, and a quick checklist of all resources that could be used on the scene to prevent injuries. This type of detailed form for known hazardous events could also be utilized in healthcare. Finally, simulation educators may be motivated to cross the healthcare line by receiving grant funding for completing similar deliverables. This funding can be used to acquire VR equipment or to pay for release time of simulation educators.

*4.3. Limitations*

This pilot was limited in several ways. The group size was restricted to four active participants and four observing participants. This limitation was due to limited participant availability for offsite training. Future trainings will be conducted with new employees with fewer onsite responsibilities to ensure a higher rate of participation. Additionally, all participants scored their pre-confidence survey items as "I am confident". In the future, facilitators should perhaps create a more psychologically safe learning environment for participants that reassures and empowers an environment of taking risks and being vulnerable in the presence of their supervisors. It is possible that the pre-confidence data could be under reported.

## 5. Conclusions

This pilot project demonstrated the successful implementation of a VR simulation in the unique context of utility lineman. There are a multitude of considerations when creating and implementing a VR simulation, yet there are established standards in simulation that may be used to guide the process. The steps revealed in this article of utilizing Kotter's (1996) Change Theory and the INACSL Standards of Best Practice: Simulation$^{SM}$ Simulation Design can assist simulation educators to have a broader scope throughout the community in implementing the design, development, and implementation of a VR simulation as an experiential learning modality to train workers in both healthcare and the community. This process may be helpful to guide or inspire further adoption of VR in unconventional settings.

**Author Contributions:** Conceptualization, A.H.; methodology, A.H.; validation, A.H. and J.T.; formal analysis, J.T.; investigation, A.H.; resources, A.H.; data curation, A.H.; writing—original draft preparation, A.H.; writing—review and editing, A.H. and J.T.; visualization, A.H.; supervision, A.H.; project administration, A.H.; funding acquisition, A.H. All authors have read and agreed to the published version of the manuscript.

**Funding:** This pilot is anticipating funding by the Energy and Natural Resources Division of the Mississippi Development Authority (ENRD).

**Acknowledgments:** The authors would like to thank and acknowledge Tonya Rutherford-Hemming for her feedback assistance in writing this article.

**Conflicts of Interest:** The authors declare no conflict of interest. The funder had no role in the design of the study; in the collection, analyses, or interpretation of data; in the writing of the manuscript; or in the decision to publish the results.

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
