# Peer review of "Crossing the Power Line: Using Virtual Simulation to Prepare the First Responders of Utility Linemen"

_informatics, doi:10.3390/informatics7030026_

Round 1

Reviewer 1 Report

This article examines the implementation and effectiveness of a VR training application for linemen workers, in a pilot study with four participants. The topic is interesting, but I have two significant concerns regarding it, which prevent me to recommend its publication. First, the sample of participants is very small. It is understandable since it is a pilot, but, for the same reason, I think this would fit better a conference presentation/article than a full article in a journal. Second, very few details about the research (method, analyses) are provided; it makes it very hard to understand what was done with enough detail. This critically limits the relevance of the research for the field.  

In general, the Method section is too vague and does not allow the reader to clearly understand what was done in the study. 

"The learning gap was identified by interviewing multiple energy 70 experts from across the state." Please be more specific: how many experts were interviewed? who conducted the interviews? which approach was followed to conduct the interviews and to analyze the results?

The description of the tasks performed by the participants is also too vague, it is hard to understand in detail how were the tasks the participants performed. 

More details of the sample of participants should be provided (age, gender, at least)

The details of the evaluation tools applied should be included in the Method section, instead of in the Results section. More details should be provided on the tools used for the evaluation (references to the surveys used, number of questions/scales/items, etc.). 

The results section is quite incomplete. Data on mean scores, standard deviations, some tests of statistical significance of the results, etc. need to be included to make the results meaningful.  

Without a more nuanced description of the method and results, it is not possible to examine whether the claims made in the discussion and conclusions are well supported by the data

Author Response

Thank you so much for your review.  I believe the suggestions you had for my article will improve the quality of this manuscript for your readers. I appreciate your consideration and time reviewing my revisions.  Please see the attachment.

Reviewer 2 Report

Authors present the results of a small, exploratory pilot study to trial the use of the International Nursing Association for Clinical Simulation and Learning (INACSL) Standards of Best Practice: SimulationSM Simulation Design with nonhealthcare learners.

The approach is interesting, however from my point of view the limited number of participants in the trial is not enough for a statistical analysis.

I recommend increasing the number of participants before accepting the paper.

Author Response

(The authors gave the same response as above.)

Reviewer 3 Report

Hi I have a attached a file with my comments.

Author Response

(The authors gave the same response as above.)

Round 2

Reviewer 1 Report

My impression is that the manuscript is improved compared to the previous version. However, I still think that the small sample size of four participants (since the observers cannot be considered, in my opinion, as participants) remains as a critical limitation. Moreover, the lack of a control group, and the fact that all measures were self-reported (questionnaires) and that no performance measures were included in the study, makes the results very sensitive to biases (e.g. social desirability bias). Hence, I think that the quality of the results is still very disputable, and this prevent me to recommend the publication of the manuscript.

Author Response

Hello,

I clarified in the paper that this pilot was a project aimed to translate existing nursing evidence of effectively using the INACSL Standards of Simulation Best Practices with nonhealthcare workers and conducting a program evaluation.  I hope this clarification helps clarify that this was not a research study.  I believe this paper is important for your readers to be able to replicate these steps to improve their learners' knowledge, skills, and attitudes in any field using virtual simulations.  

Thanks for your consideration,

Alaina

Reviewer 2 Report

The paper is well organized, the state of the art should be improved, there are only 11 references.

The number of participants for the testing sessions is still low.

Author Response

(The authors gave the same response as above.)

Reviewer 3 Report

Article Entitled: Exploring Steps and Benefits of crossing the “Healthcare Power Line”: Using Virtual Simulation Training to Prepare the Forgotten First Responders

Thank you for allowing me to review this revised article entitled “Exploring Steps and Benefits of crossing the “Healthcare Power Line”: Using Virtual Simulation Training to Prepare the Forgotten First Responders” sent to the Informatics Journal. Your submission changes were completed well and sufficiently address all of my comments. The article will provide readers with excellent example of how we can share simulation learnings to other disciplines.

I would accept this article for publication after being reviewed by an editor for minor grammatical errors.

Author Response

Hello,

Thank you for your valuable feedback. I believe this paper is important for your readers to be able to replicate these steps to improve their learners' knowledge, skills, and attitudes in any field using virtual simulations.  

Thanks for your consideration,

Alaina